# Batten Down the Hatches—Assessing the Status of Emergency Preparedness Planning in the German Water Supply Sector with Statistical and Expert-Based Weighting

**Lisa Bross** [1,*] **, Ina Wienand** [2] **and Steffen Krause** [1]

1   Research Center RISK, Bundeswehr University Munich, Werner-Heisenberg-Weg 39,
    85577 Neubiberg, Germany; steffen.krause@unibw.de
2   Federal Office of Civil Protection and Disaster Assistance, Provinzialstrasse 93, 53127 Bonn, Germany;
    ina.wienand@bbk.bund.de
*   Correspondence: lisa.bross@unibw.de

**Abstract:** Emergency preparedness planning in the water supply sector includes preventive measures to minimize risks as well as aspects of crisis management. Various scenarios such as floods, power failures or even a pandemic should be considered. This article presents a newly developed composite indicator system to assess the status of emergency preparedness planning in the German water supply. Two weighting methods of the indicators are compared: the indicator system was applied to a case study and a Germany-representative data set. The results show that there is a need for action in emergency preparedness planning in the German water supply. This is in particular due to a lack of risk analyses and insufficient crisis management. Numerous water supply companies and municipalities are already well-prepared, however, there is a need for action at several levels, especially in the area of risk analysis and evaluation of measures. In Germany, responsibility for this lies primarily with the municipalities.

**Keywords:** crisis; critical infrastructure; disaster; drinking water; risk management; risk reduction

## 1. Introduction

The corona pandemic in 2020 affects society, as well as essential services such as the water utilities, in a new and profound way [1]. Critical infrastructures like water utilities and the provision of their vital services in such a scenario have an outstanding importance to a nation's society. Their failure or degradation could result in sustained supply shortages, which affect public health, economy and national security. Quarantined personnel, working from home to distance employees and unpredictable supply chains for consumables are new and unfamiliar conditions that make the operation of water supply companies more difficult and potentially endangers overall water supply [2].

A resilient drinking water supply is consequently one of the basic requirements for a stable social and economic system. However, impairments cannot be completely avoided, so that water supply companies have to deviate from normal operation, e.g., in the event of pipe bursts. Such minor disturbances occur comparatively frequently and have only minor effects [3]. They can usually be quickly identified and repaired. As a consequence they usually remain unnoticed to the consumers [4]. On the contrary, failures or more extensive impairments of the water supply systems can have serious impacts on the affected population and the economy [3,5–7]. Causes can be serious natural events, man-made accidents or intentional attacks [8,9], whose probability of occurrence is constantly increasing [10].

The corona pandemic puts the importance of critical infrastructures, and the need for proactive emergency preparedness planning to increase their resilience, at the forefront of civil protection and disaster management [1]. The understanding, analysis and quantification of resilience by water utilities, authorities, decision-makers and other stakeholders is a prerequisite for this.

The resilience of water supply systems can be increased by appropriate emergency preparedness planning instead of ad hoc coping responses. This includes the conceptual, organisational and technical preconditions for risk reduction and prepares structures for response in the event of a crisis [11]. Emergency preparedness planning in the water supply sector thus comprises, in addition to measures to avoid damaging events, especially preventive, safeguarding, reactive and restorative aspects of risk and crisis management.

Effective emergency preparedness planning is characterised, among other things, by the fact that the planning is carried out as preventive measures and the measures can be implemented in emergency situations. Beyond preventive measures to minimise risks, emergency preparedness planning in the water supply sector includes in particular aspects of crisis management [12]. Such emergency preparedness planning takes into account different scenarios and their possible effects on the water supply. In addition to preventive measures, the numerous aspects of crisis management also lead to risk minimization by limiting the extent of damage. Figure 1 shows the five steps of risk and crisis management according to the German Federal Ministry of the Interior [13] and the Federal Office of Civil Protection and Disaster Assistance [12,14].

Thorough preliminary planning forms as the first process step the basis for the successful establishment of risk and crisis management [13]. Basic specifications should be made in advance of the establishment or expansion of a risk and crisis management system. These include the promotion of risk awareness and the definition of key players as well as responsibilities in the course of the emergency preparedness planning process [14].

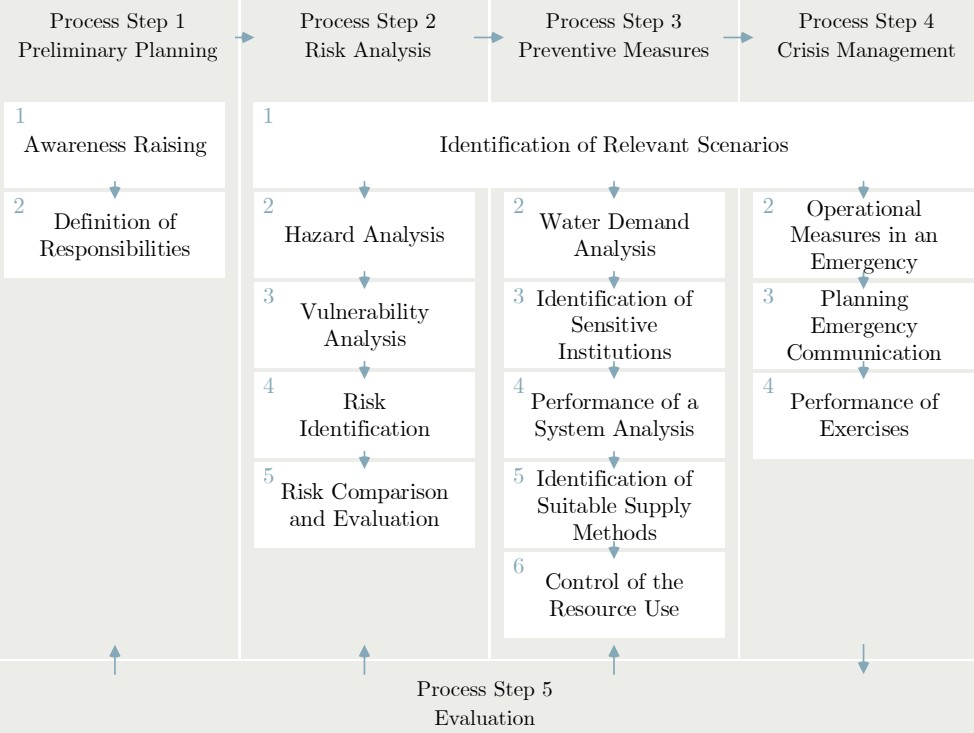

**Figure 1.** Structuring of the procedure for emergency preparedness planning in water supply based on [12,13,15].

A risk analysis structures and objectifies the collection of information on existing and potential risks to the water supply [14]. The analysis considers the reasons and causes of risks, examines

the possible effects and determines the framework within which these consequences can occur [16]. In addition, risk analysis provides the basis for effective and efficient use of limited resources by comparing the various identified risks of processes and components of water supply.

Preventive measures contribute to the reduction of risks for critical processes. They also contribute to achieve operational protection goals and thus raise the barrier for events with crisis potential in the facility [12]. In this way, the number of crisis-prone events can be minimized or the intensity of the events can be reduced.

The processes of crisis management help to protect facilities and thus critical infrastructures and the population. Interactions exist with risk management, since not all risks can be reduced by risk-minimizing measures and a residual risk always remains [12]. Crisis management therefore offers a structure for coping with crises that cannot be prevented [13,17].

The evaluation refers to all phases, i.e., both the examination of points defined in the preliminary planning, the examination of the topicality of the information on existing risks, the examination of the effectiveness of the implemented preventive measures and the examination of the crisis management [14]. It should be repeated regularly.

## 2. Materials and Methods

The emergency preparedness planning indicator (EPP) developed in this study is based on a number of main, partial and individual indicators. These indicators cover organisational as well as technical aspects of emergency preparedness planning. The contents of the indicator correspond to the processes and components of effective emergency preparedness planning in water supply. For the development and calculation of the EPP, a multi-stage and iterative process according to [18] was carried out (Figure 2).

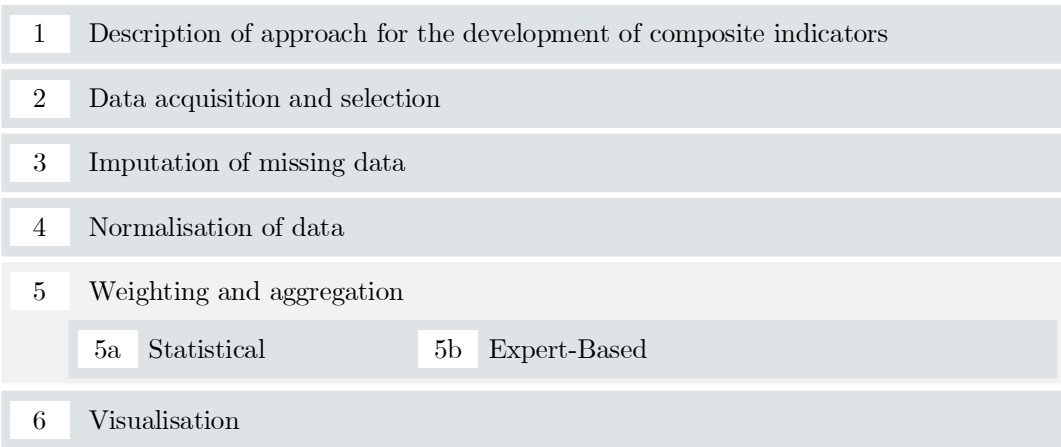

**Figure 2.** Procedure for the compilation of a composite indicator based on [18].

### 2.1. Description of Approach for the Development of Composite Indicators

The theoretical framework of the EPP is the systematic procedure of risk and crisis management according to [13,14], the necessary scope of which is described by five process steps. The EPP therefore consists of the five main indicators of preliminary planning, risk analysis, preventive measures, crisis management and evaluation. The Table 1 shows the subdivision of the five main indicators and the 19 sub-indicators in total. Appendix A Tables A1–A7 show all individual indicators. In addition to a literature study and existing theoretical models, the individual indicators representative for emergency preparedness planning were developed on the basis of expert and stakeholder knowledge in different workshops, thus applying a stakeholder-oriented methodology. This methodology is usually used in the development of composite indicators (e.g., [19,20]), if, as in the present case, their use is intended as a self-assessment tool for municipalities or authorities.

**Table 1.** Composition and hierarchy of the emergency preparedness planning indicator through five main indicators as well as their sub and individual indicators.

| | Main Indicator | | Sub Indicator | Individual Indicator | Source | In NoWa I Dataset |
|---|---|---|---|---|---|---|
| PP | Preliminary Planning | $PP_1$ | awareness raising | $PP_{1,1}$ to $PP_{1,9}$ | [12,14,21,22] | Yes |
| | | $PP_2$ | definition of responsibilities | $PP_{2,1}$ to $PP_{2,3}$ | [14,21,23] | No |
| RA | Risk Analysis | $RA_1$ | hazard analysis | $RA_{1,1}$ to $RA_{1,8}$ | [14,17,24] | Yes |
| | | $RA_2$ | vulnerability analysis | $RA_{2,1}$ to $RA_{2,8}$ | [14,17] | No |
| | | $RA_3$ | risk identification | $RA_{3,1}$ to $RA_{3,7}$ | [14,16,17,25] | Yes |
| | | $RA_4$ | risk comparison and assessment | $RA_{4,1}$ to $RA_{4,2}$ | [14] | No |
| PM | Preventive Measures | $PM_1$ | structural redundancies | $PM_{1,1}$ to $PM_{1,2}$ | [14,17] | Yes |
| | | $PM_2$ | interrelation of supply | $PM_{2,1}$ | [14,17,26] | Yes |
| | | $PM_3$ | grid construction | $PM_{3,1}$ to $PM_{3,2}$ | [17,27,28] | No |
| | | $PM_4$ | remote monitoring, control systems | $PM_{4,1}$ to $PM_{4,2}$ | [14,17,29,30] | No |
| | | $PM_5$ | general measures | $PM_{5,1}$ to $PM_{5,3}$ | [17,31,32] | Yes |
| CM | Crisis Management | $CM_1$ | organisation and coordination | $CM_{1,1}$ to $CM_{1,7}$ | [12,14,17,23,33–35] | Yes |
| | | $CM_2$ | provision of resources | $CM_{2,1}$ to $CM_{2,2}$ | [12,14] | Yes |
| | | $CM_3$ | exercises | $CM_{3,1}$ to $CM_{3,2}$ | [12,14,17,23] | Yes |
| | | $CM_4$ | communication | $CM_{4,1}$ | [14,17,36,37] | Yes |
| E | Evaluation | $E_1$ | evaluation preliminary planning | $E_{1,1}$ to $E_{1,2}$ | [14,17] | No |
| | | $E_2$ | evaluation risk analysis | $E_{2,1}$ to $E_{2,4}$ | [14,17,38] | Yes |
| | | $E_3$ | evaluation preventive measures | $E_{3,1}$ to $E_{3,5}$ | [14] | No |
| | | $E_4$ | evaluation crisis management | $E_{4,1}$ to $E_{4,4}$ | [14,17,23,36] | Yes |

## 2.2. Data Acquisition and Selection

The indicators shown in Table 1 are necessary for the quantitative assessment of the status of implementation of emergency preparedness planning. Thus, data are required that allow both a review of the applicability and significance of the EPP and the determination of the status quo in Germany. However, the required information cannot be determined from publicly available data, as this is utility-specific and relevant to the security of the water utility and its services. A targeted assessment is therefore necessary.

### 2.2.1. Case Study for Verification of Indicator

To check the applicability and significance of the EPP, all indicators were collected on the basis of a questionnaire for a water supply company as a case study. The case study shows a real water supply utility which supplies a total of 230,000 inhabitants in 120 municipalities and districts. In total, the water supply company delivers about 10 million cubic meters of water annually.

### 2.2.2. Germany-Representative Dataset

In order to assess the status quo of emergency preparedness planning in Germany, an existing dataset of a nationwide survey on emergency preparedness planning in water supply was analysed. The data set was collected in 2015 by means of a partially standardised questionnaire within the framework of the NoWa I research project with the assistance of the federal level in order to obtain a general, supra-regional overview of the current status of emergency planning in the districts and municipalities. As the responsible bodies, the districts and municipalities had to ask for input from the water supply companies to fill out the data collection form, if the information was not already available. In total, a completed survey questionnaire was returned by 194 districts and 166 municipalities. The data thus consists of 360 individual data sets, which contain data from nationwide distributed municipalities and districts with a population of around 39 million inhabitants.

Each dataset contains information on 37 questions concerning different aspects of emergency preparedness planning and existing water supply systems. To determine the status quo, 21 relevant individual indicators from the data entry form with 37 questions were identified and considered in the EPP. In total, the data sets are assigned to twelve of the 19 sub-indicators (Table 1).

A subsequent data collection or data supplement could not be implemented, since the data collection within the framework of the NoWa I project and an additional collection of a representative data set could not be repeated. Thus, the determination of the status quo does not include all identified indicators.

## 2.3. Imputation of Missing Data

Missing data impairs the development and evaluation of composite indicators and can lead to a distortion of the results [18]. In the present study, a case-by-case elimination of data sets is only applied if the data that are absolutely necessary for the situation analysis (e.g., allocation of the data set to the municipality) are not available.

The analysed data of the NoWa I project were collected before the methodology of the emergency preparedness planning indicator was developed. They do not include all indicators relevant to the EPP. Since it was not possible to collect such a data set subsequently, the NoWa I data sets were used to determine the indicator, although they did not include complete indicator data sets (as shown in Table 1). The missing individual indicators are therefore not included in the evaluation.

## 2.4. Normalisation of Data

In order to be able to compare the indicators of different municipalities or the individual sub-indicators with each other, a normalisation process is necessary. This is especially true if the data sets differ in their units of measurement [18].

The questions with "yes-no" or "yes-partial-no" possible answers are converted into a [0,1] scale. Likert scales with a given answer scale were also transformed into a [0,1] scale. The answer option "not known" was equated to the answer option "no", since this is equivalent in terms of content for the evaluation of the indicator. This was necessary because the originally planned survey in the NoWa I project had a different assignment of the questions.

### 2.5. Weigthing and Aggregation

The individual indicators are integrated into the composite indicator with different weightings (see Equation (1)). This is because the composite indicator is calculated by the weighted sum of its main indicators (see Equation (2)). For the contingency planning indicator, the weighted sum of the five main indicators is determined. As described in [39], there are several ways to determine the weighting of composite indicators.

$$CI = \sum_{j=1}^{m} x_j X_j \tag{1}$$

$$EPP = x_{PP} \cdot PP + x_{RA} \cdot RA + x_{PM} \cdot PM + x_{CM} \cdot CM + x_E \cdot E \tag{2}$$

$CI$　　Composite indicator
$m$　　Number of main indicators
$x_j$　　weight of main indicator j
$X_j$　　normalized value of the main indicator (PP, RA, PM, CM and E)

For the EPP this paper compares the results of a statistical and an expert-based weighting approach. The main difference between the two approaches is how the indicator weights are derived. Since the weighting of the main and sub-indicators significantly influences the result of the EPP, the composite indicator is determined with identical main and sub-indicators for both weighting approaches.

If the main indicators are equally weighted, they are equally included in the composite indicator. Due to the different weighting of the individual summands (main indicators or sub-indicators), they are assigned a differentiated significance for the composite indicator.

To determine the expert-based weighting, the weights were derived from expert opinions. Using a budget allocation approach, fourteen experts with different specialist backgrounds were asked to assess the main and sub-indicators in a questionnaire using a Likert scale according to their relevance for target-oriented emergency preparedness planning. The weightings of the main indicators derived from the equal distribution and from the expert opinions are presented in Section 3.1.

### 2.6. Visualisation

The results of the composite and main indicators are visualized in an anonymized representation using treemap diagrams. This type of presentation was chosen because the data set depicts a large number of municipalities and water supply utilities, but the number of inhabitants varies greatly. The hierarchical structure of the treemap diagrams makes the proportions of the municipalities and water supply companies under consideration clear. At the same time all results are visible. Rectangles of different sizes are used to display the number of inhabitants of the district or the district-free city or municipalities ($E_i$) in relation to the inhabitants of the entire data set ($E_{ges}$) (see Figure 3). Thus, each rectangle corresponds to a municipality, which is always at the same position in the respective diagrams. The color of the rectangles represents the value of the indicator. The evaluation of the data sets is carried out anonymously. In addition, the possibilities of drawing conclusions about individual municipalities are minimized by the following selected representation.

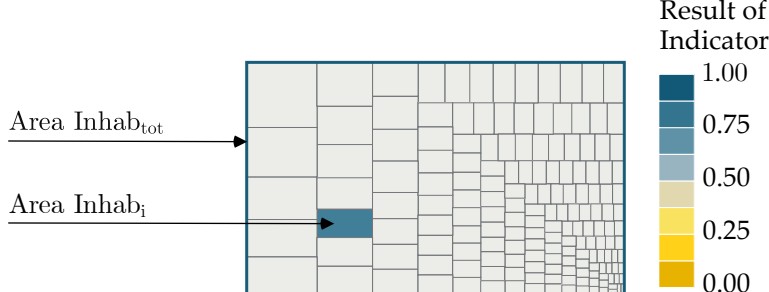

**Figure 3.** Explanation of the visualization of the analysis of the status of emergency preparedness planning through treemap diagrams.

## 3. Results

### 3.1. Determination of the Weighting of the Indicators

Emergency preparedness planning comprises the five process steps preliminary planning (1), risk analysis (2), preventive measures (3), crisis management (4) and evaluation (5). In case of equal distribution, the weighting of the five process steps corresponds to 20% or $x_j = 0,2$ each. The expert-based weights are between 17 and 22% (Figure 4). The mean value of the expert-based weight of the process step Preliminary Planning (1) is 0.22, the highest value, and the mean value of the expert-based weight of the process step Evaluation (5) is 0.17, the lowest value of the five weight. The expert opinion regarding the weight of the process step Evaluation (5) varies the strongest. The difference in the mean values of the expert-based weightings is statistically significant ($p < 0.05$).

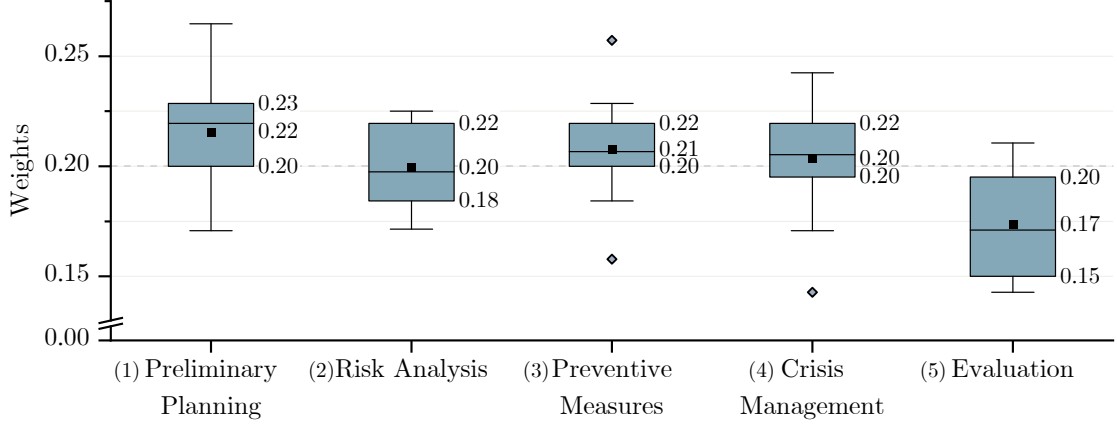

**Figure 4.** Expert-based weighting of the five process steps of emergency preparedness planning ($n = 14$).

### 3.2. Assessment of the Applicability and Significance of the Emergency Preparedness Planning Indicator Based on the Case Study

In order to determine the status quo of emergency preparedness planning, a data collection form was compiled from the indicators listed. This was answered by the water supply company and the responsible disaster control authority from the case study and the data was evaluated.

For the case study, this results in an $EPP_S$ of 0.66 and an $EPP_E$ of 0.67 (Figure 5). The main indicator PP with a value of 0.96 corresponds to the highest result of the five main indicators. The lowest value is obtained for the main indicator CM with 0.43 and 0.44. This means that in the area of preliminary planning almost all aspects have been implemented in the company and district, but in the area of preventive crisis management aspects are not yet sufficiently practiced.

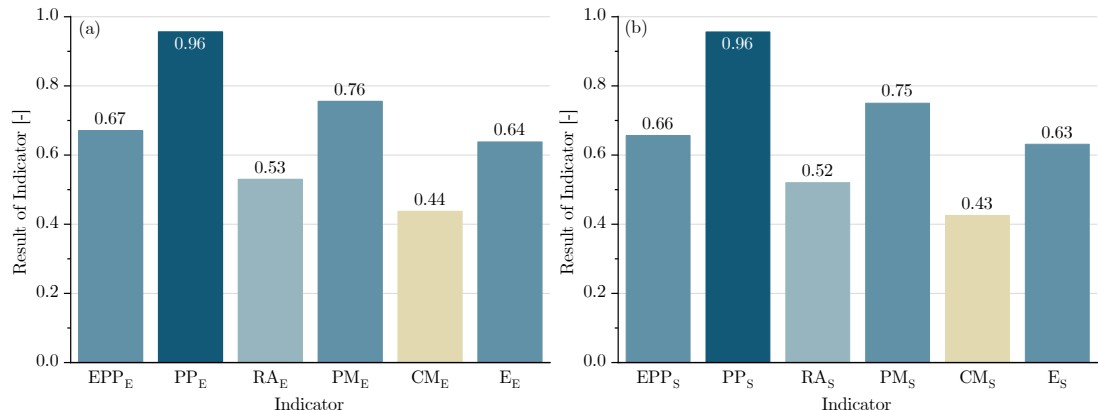

**Figure 5.** Results of the EPP and the main indicators for the case study based on the different weightings (**a**) Expert-based Weights (**b**) Equal Weights.

The methodology developed in this study to assess the status of emergency preparedness planning is applicable in practice. The applied questionnaire captures the relevant aspects for assessing the status quo and deriving the need for action. The quantitative results also enable a comparison between municipalities and the prioritisation of planned measures to address the need for action.

### 3.3. Status of Emergency Preparedness Planning in Germany

#### 3.3.1. Data Basis for the Analysis of the Status Quo of Emergency Preparedness Planning

In order to determine the status of emergency preparedness planning in the water supply sector, a total of 360 data sets were analysed and evaluated. These comprise 194 data sets from districts and 166 data sets at the municipal level. Due to the different responsibilities in Germany, the data sets of the districts and municipalities are evaluated separately from the data sets at the municipal level (Table 2). The data sets are each divided into four groups of approximately equal size according to the number of inhabitants covered. In addition, a distinction was made between the areas of responsibility of the senders of the data collection forms. It is therefore necessary to examine, whether differences can be identified with regard to the level of preparation of the different senders and the size of the municipality.

**Table 2.** Size of the municipalities and districts as well as the field of activity of the senders of the survey forms.

| Inhabitants | Proportion of Survey Forms | | | | | |
|---|---|---|---|---|---|---|
| | Water Utility | Administration | Civil Protection | Health Department | Multiple Senders | Line Sum |
| Municipalities (*n* = 166) | | | | | | |
| to 3.000 | 7% | 20% | 0% | 0% | 0% | 27% |
| 3.001 to 5.000 | 4% | 16% | 0% | 0% | 0% | 20% |
| 5.001 to 10.000 | 11% | 14% | 0% | 0% | 0% | 25% |
| more than 10.000 | 19% | 3% | 0% | 3% | 3% | 28% |
| Column Sum | 41% | 53% | 3% | 0% | 3% | 100% |

**Table 2.** *Cont.*

| Inhabitants | Proportion of Survey Forms | | | | | |
|---|---|---|---|---|---|---|
| | Water Utility | Administration | Civil Protection | Health Department | Multiple Senders | Line Sum |
| Districs (*n* = 194) | | | | | | |
| to 100.000 | 0% | 6% | 10% | 3% | 4% | 23% |
| 100.001 to 150.000 | 0% | 7% | 12% | 4% | 2% | 25% |
| 150.001 to 250.000 | 2% | 5% | 18% | 1% | 4% | 30% |
| more than 250.000 | 2% | 1% | 11% | 6% | 2% | 22% |
| Column Sum | 4% | 19% | 51% | 14% | 12% | 100% |

### 3.4. Assessment of the Status of Emergency Preparedness Planning in Germany

In order to assess the status of the emergency preparedness planning of the districts as well as the municipalities, the composite indicator EPP and the main indicators PP, RA, PM, CM and E were determined on the basis of the data sets presented. For this purpose, the results of the emergency preparedness planning indicators with weighting according to expert opinion ($EPP_E$) and with equally distributed weighting ($EPP_S$) are presented below.

The results of the $EPP_E$ and $EPP_S$ vary for the districts in a few cases (Figure 6) The mean value of the $EPP_E$ as well as the $EPP_S$ is equal to 0.42 (Table 3). The small differences in the weighting show only a little effect in the result. Moreover, these differences are not statistically significant ($p > 0.05$).

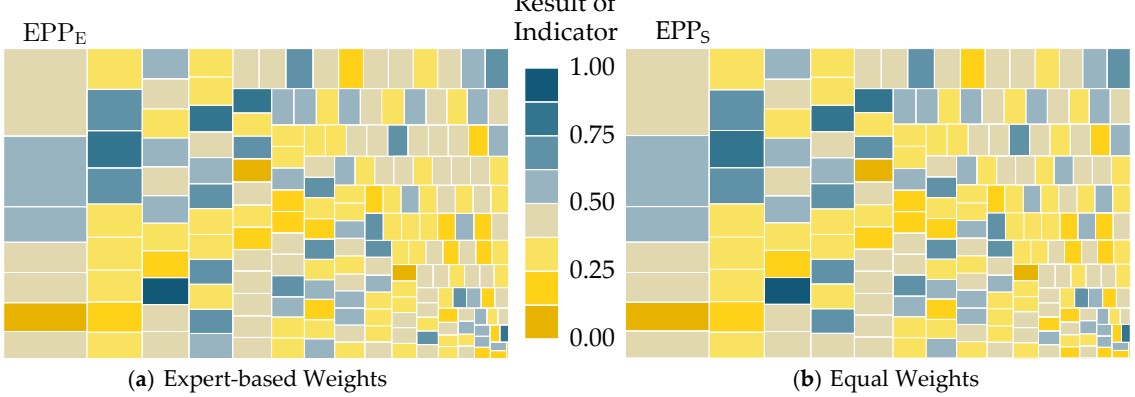

**Figure 6.** Emergency preparedness planning indicator according to the size of the districts.

**Table 3.** Results of the emergency preparedness planning indicator with expert based and equally distributed weighting and the main indicators.

| | Indicator | Mean Value of Indicator (Standard Derivation) | |
|---|---|---|---|
| | | Districs and Municipalities | |
| EPP | $EPP_E$ | 0.42 (SD = 0.17) | 0.32 (SD = 0.18) |
| | $EPP_S$ | 0.42 (SD = 0.16) | 0.32 (SD = 0.18) |
| Main Indicator | PP | 0.43 (SD = 0.25) | 0.30 (SD = 0.27) |
| | RA | 0.16 (SD = 0.32) | 0.14 (SD = 0.29) |
| | PM | 0.57 (SD = 0.31) | 0.39 (SD = 0.38) |
| | CM | 0.55 (SD = 0.32) | 0.43 (SD = 0.22) |
| | E | 0.38 (SD = 0.22) | 0.31 (SD = 0.25) |

The participating districts show a different level of preparation. Districts with more inhabitants achieve a higher EPP (Figure 7). The range of EPP increases with the number of inhabitants, so that the largest districts show the greatest difference in the level of emergency preparedness within a size group. The differences in the mean values by size of the districts are significant in both cases ($p < 0.05$).

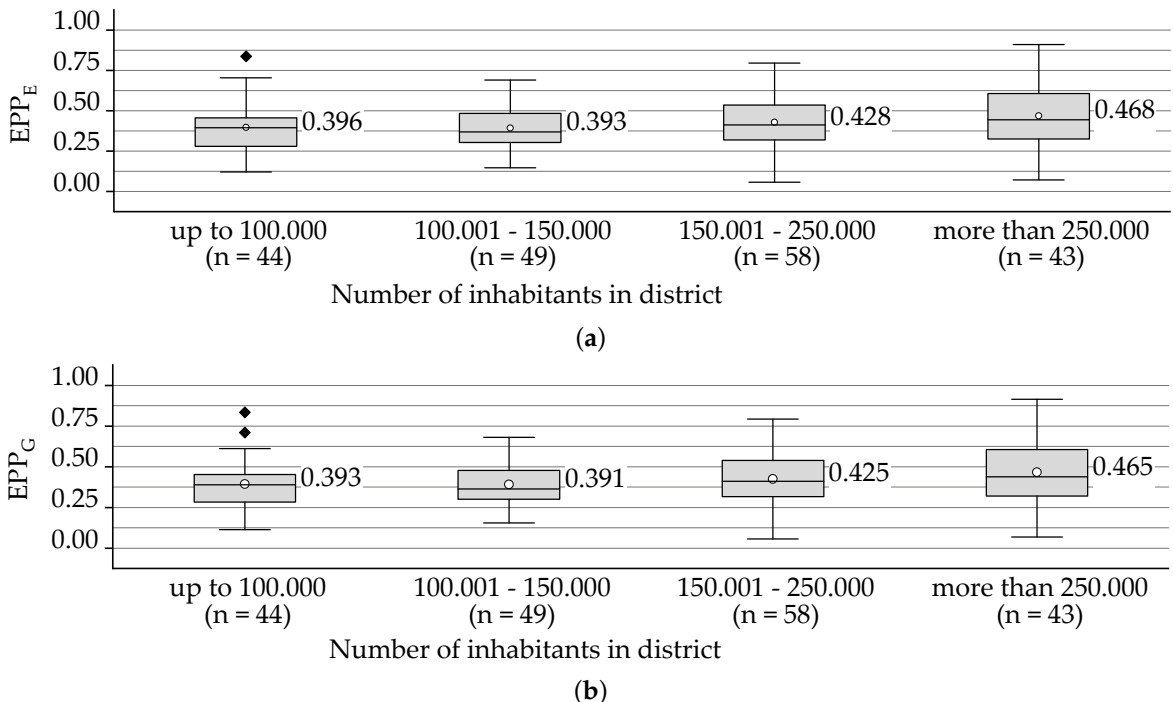

**Figure 7.** EPP according to the size of the districts. (**a**) Expert-based weighting of the main and sub-indicators; (**b**) Equally distributed weighting of the main and sub-indicators.

The mean values of the $EPP_E$ and $EPP_S$ for municipalities are 0.32. Some municipalities have thus already implemented certain aspects of emergency preparedness planning. However, these implementations are still in their beginnings. Differences between the $EPP_E$ and $EPP_S$ are only evident in a few cases (Figure 8), but do not show statistical significance ($p > 0.05$).

Municipalities with an increasing number of inhabitants achieve higher results in the $EPP_E$ as well as in the $EPP_S$. The differences in the mean values between the size of the municipalities are significant ($p < 0.05$). However, the range of the $EPP_E$ does not increase with a growing number of inhabitants in the municipality, which can be seen at the level of districts.

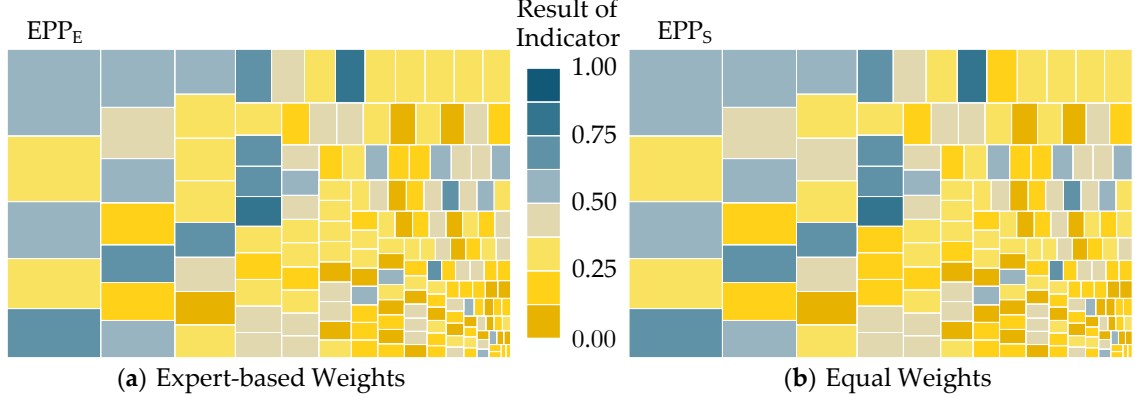

**Figure 8.** Emergency preparedness planning indicator by size of municipalities.

The Figure 9 shows the differences of the $EPP_E$ and $EPP_S$ between expert-based and equally distributed weighting. The colored variance of the rectangles shows that the different weightings affect the results of the $EPP_E$ and $EPP_S$. In addition, the figure shows that the difference between $EPP_E$ and $EPP_S$ is in the range of $\pm 0.018$ and is therefore only noticeable in a few cases when the result is rounded to the second decimal place.

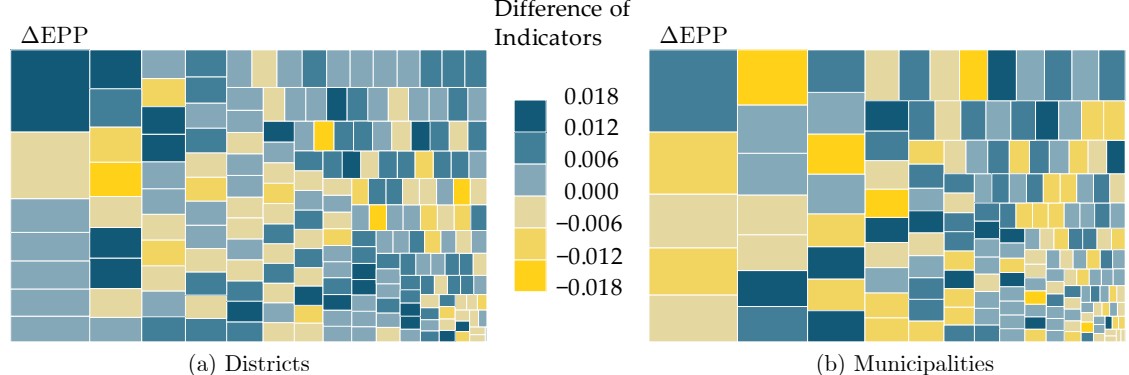

**Figure 9.** Difference between the emergency preparedness planning indicators $EPP_E$ and $EPP_S$ according to the size of the data sets taken into account.

## 4. Discussion

### 4.1. Discussion of the Applied Methods

For effective planning and implementing of measures to increase the resilience of water supply systems, it is necessary to assess the status of emergency preparedness. The Emergency Preparedness Planning Indicator was developed to increase transparency by quantifying this assessment. The EPP was tested on the basis of the case study for its applicability and significance. The determination of the status of emergency preparedness planning was implemented by means of a data set of the research project NoWa I which is representative for Germany.

The composite indicator developed in this study enables a measurable assessment of the status of emergency preparedness planning. The main and sub indicators included are based on scientific publications (e.g., [12,14,17,23]), so that they reflect the current state of research on emergency preparedness planning. The development and application of the composite indicator is based on two main motivations. Firstly, the indicator can serve as a tool for self-assessment of emergency preparedness planning by asking specific questions. The self-assessment can be used by water utilities, regional or national authorities and municipalities to improve the prevention measures. The result of the indicator can also raise awareness of the relevance of emergency preparedness planning and show the need for action.

Secondly, the indicator supports local and regional authorities as well as national and international organisations in the assessment process, by comparing different municipalities, e.g., in the context of benchmarking processes of water supply systems. The results should support these institutions in decision-making, e.g., on the allocation of resources, and make them more transparent and consistent.

Two weighting methods were used to determine the emergency preparedness planning indicator. An equally distributed weighting of indicators is the standard assumption in the literature (e.g., [40,41]). If equal weightings are used, the indicators are either constructed in such a way that each variable or branch of a hierarchy level is equally weighted. To determine the EPP, equal weighting was applied to each hierarchical level. Furthermore, a weighting of the main and sub-indicators was applied based on expert knowledge. However, no strict participatory method has been applied here, where all weightings included are based on expert opinion [39]. This was only used for the weighting of the main and sub-indicators, but not for the individual indicators. For the individual indicators, aggregation at each level by arithmetic means was applied. Due to the different number of partial and individual

indicators, they have a different weight despite their apparent equal weighting [42]. This also applies to the evaluated data set of the NoWa I project and must be taken into account when deriving the need for action.

The results show only minor differences with the two different weighting methods, both for the data set of the NoWa I project and the case study. Consequently the experts' opinions confirm the relevance of all five process steps of risk and crisis management according to [13,14]. In order to increase the acceptance of the indicator and to simplify the calculation, it is therefore recommended that the weighting be applied by using statistically equally distributed weights.

### 4.2. Discussion of the Indicator and Assessment Results

The results of the status of emergency preparedness planning for the case study show that some measures have already been implemented. An advanced state of implementation can be seen, particularly in the preliminary planning (1) and preventive measures (3). However, since the status of the risk analysis (2) still needs to be improved, the appropriate identification and implementation of preventive measures cannot be in a targeted manner. For this reason, the evaluation of the preventive measures (3) is necessary following the complete implementation of the risk analysis (2). Further action is needed in the area of crisis management (4). The sub-indicator results of this process step refer to the lowest level of implementation in the case study. However, since the crisis management (4) measures are based on the risk analysis (2) and the preventive measures (3), the implementation of a systematic approach step-by-step is an important prerequisite. Nevertheless, the individual process steps must still be evaluated and updated regularly.

The results of the composite indicator show a very heterogeneous picture with regard to the implementation status of emergency preparedness planning in the German water supply sector. The heterogeneity of the emergency preparedness planning indicator exists both in the districts and at the municipal level. Furthermore, the results of the respective five process steps are very diverse. Thus, some aspects of emergency preparedness planning have already been implemented. Nevertheless, implementation is still insufficient in some districts and municipalities.

The differentiation of districts or municipalities according to their number of inhabitants indicates that in both cases, larger municipalities have on average better emergency preparedness planning than smaller ones. Nevertheless, both groups contain outliers in both directions. Reasons for a more sophisticated emergency preparedness planning in larger municipalities may be, on the one hand, an extensive staffing or optimized structural conditions. On the other hand, the water supply in larger municipalities in Germany often lies in the responsibility of larger supply utilities, which often devote themselves more intensively to this task due to their corporate structure.

Indicator methods can reduce their usefulness for policy-makers or even lead to disadvantageous decision-making due to over-simplification of complex concepts and the use of aggregation procedures that are difficult to understand [43]. Although the answers to the questions used to determine the EPP are checklist-based self-assessments, the scope of the questions could lead to little attention being paid to the individual answers. The result of the case study shows that the procedure is applicable and the self-assessment questionnaire with 74 questions is appropriate. A shorter list of questions would lead to a less meaningful result.

The emergency preparedness planning indicator enables a quantitative comparison between municipalities or water supply utilities as well as the five process steps and the identification of the need for action. In the German water supply sector, the implementation of risk analysis should be emphasized. Preventive measures that have already been taken have to be checked for their appropriateness following the successful implementation of a risk analysis. Threshold values for sufficient or improvable emergency preparedness planning have to be defined. However, this cannot be achieved by science alone. This requires in particular a discourse between scientists, technical experts and political decision-makers.



### 5. Conclusions

Effective emergency preparedness planning is characterised, among other things, by the fact that its process steps can be carried out with foresight and the defined measures can be effectively put into practice in emergency situations. The five process steps of risk and crisis management have to be systematically taken into account.

The status of emergency preparedness planning in Germany was determined using a composite indicator. The data basis of the indicator system is formed by the case study of a water supply company and a survey of the NoWa I research project, which is representative for Germany. The results indicate a need for action in the different processes of emergency preparedness planning, because the process steps (1–5), especially risk analysis (2) are carried out rarely or insufficiently. However in the area of preliminary planning (1), numerous water supply companies and municipalities are already well positioned, and several preventive measures (3) are also being implemented. The regular evaluation (5) of these measures could be improved.

A need for action is especially identified in the development of practicable tools for implementation of an integrated risk and crisis management process in order to intensify the exchange of the relevant actors. Furthermore training in the area of risk and crisis management with the emphasis on extraordinary or extreme events should be conducted by water utilities and local authorities.

**Author Contributions:** Conceptualization: L.B., S.K., and I.W.; methodology: L.B. validation: S.K. and I.W.; formal analysis: S.K. and I.W.; investigation: L.B., I.W., and S.K.; resources: S.K. and I.W.; data curation: L.B.; writing—original draft preparation: L.B.; writing—review and editing: L.B., S.K., I.W.; visualization: L.B.; supervision: S.K. and I.W.; project administration: S.K. and I.W.; funding acquisition: S.K. All authors have read and agreed to the published version of the manuscript.

**Funding:** This research received no external funding.

**Acknowledgments:** The authors would like to thank all those involved in the survey for their commitment and support. Furthermore, the authors would like to thank Salomé Parra and Christian Platschek, who laid the foundation for further work through their preliminary work within the framework of the NoWa I project, as well as Renate Solmsdorf, Sybille Rupertseder and Karolina Eggersdorfer, who helped to digitise the data sets. This study was supported by the Research Center RISK and the Bundeswehr University Munich.

**Conflicts of Interest:** The authors declare no conflict of interest.

### Abbreviations

The following abbreviations are used in this manuscript:

| | |
|---|---|
| CM | Crisis Management |
| E | Evaluation |
| EPP | Emergency Preparedness Planning Indicator |
| $EPP_S$ | Emergency Preparedness Planning Indicator with statistical weights of main indicators |
| $EPP_E$ | Emergency Preparedness Planning Indicator with weights based on expert knowledge |
| PM | Preventive Measures |
| PP | Preliminary Planning |
| RA | Risk Analysis |
| E | Evaluation |

# Appendix A

**Table A1.** Survey sheet to assess the Emergency Preparedness Planning Indicator—Part 1.

| ID | Question | Reply Options | Source | Individual Indicator |
|---|---|---|---|---|
| | Main Indicator PP—Preliminary Planning | | | |
| | Sub Indicator $PP_1$—Awareness Raising | | | |
| 1 | Is it known how the supply zone is divided into municipalities or neighbourhoods? | Yes, partially, no | [14] | $VP_{1,1}$ |
| 2 | Does the possible amount of water discharge correspond to the defined protection goals? | Yes, partially, no | [14] | $VP_{1,2}$ |
| 3 | Is the technically maximum possible discharge quantity from the own extraction plant known? | Yes, partially, no | [14] | $VP_{1,3}$ |
| 4 | Is the technically maximum possible water supply from other water utilities known? | Yes, partially, no | [14] | $VP_{1,4}$ |
| 5 | Is the capacity of your own extraction plants known? | Yes, partially, no | [14] | $VP_{1,5}$ |
| 6 | Is it known which part of the supply zone is supplied by which water extraction plant? | Yes, partially, no | [14] | $VP_{1,6}$ |
| 7 | Is the following known for the individual tanks for water storage? | | | $VP_{1,7}$ |
| 7.1 | Origin of water | Yes, partially, no | [14] | $VP_{1,7,1}$ |
| 7.2 | Capacity of the storage tanks | Yes, partially, no | [14] | $VP_{1,7,2}$ |
| 7.3 | max. possible feed-in quantity from the storage tanks into the grid | Yes, partially, no | [14] | $VP_{1,7,3}$ |
| 8 | Is there an awareness in your community that there may be a quantitative impairment of the water supply? | Yes, partially, no | [14,21] | $VP_{1,8}$ |
| 9 | Is there an awareness in your community that there may be a qualitative impairment of the water supply? | Yes, partially, no | [14,21] | $VP_{1,9}$ |
| | Sub Indicator $PP_2$—Definition of Responsibilities | | | |
| 10 | Is it known who is the contact person for emergency situations in the water supply utility/ies? | Yes, partially, no | [12,14] | $VP_{2,1}$ |
| 11 | Has a crisis task force been set up? | Yes, partially, no | [12,14] | $VP_{2,2}$ |
| 12 | Is the organizational and operational structure defined? | Yes, partially, no | [12,14, 44] | $VP_{2,3}$ |

**Table A2.** Survey sheet to assess the Emergency Preparedness Planning Indicator—Part 2.

| ID | Question | Reply Options | Source | Individual Indicator |
|---|---|---|---|---|
| | Main Indicator RA—Risk Analysis | | | |
| | Sub Indicator $RA_1$—Hazard Analysis | | | |
| 13 | Has a risk analysis been carried out? | Yes, partially, no | [14,17] | $RA_{1,1}$ |
| 14 | Is there a list of which hazards have already been considered and which are not yet part of the risk analysis? | Yes, partially, no | [14] | $RA_{1,1}$ |
| 15 | From the point of view of the water supply utility, do the following exceptional events represent relevant hazards for the water supply? | | [14,24] | $RA_{1,2}$ |
| 15.1 | Natural hazards | Yes, partially, no | [14,24] | $RA_{1,2,1}$ |
| 15.2 | Accidents (human failures) | Yes, partially, no | [14,24] | $RA_{1,2,2}$ |
| 15.3 | Accidents (technical failures) | Yes, partially, no | [14,24] | $RA_{1,2,3}$ |
| 15.4 | Terrorism | Yes, partially, no | [14,24] | $RA_{1,2,4}$ |
| 16 | Does the hazard analysis include experiences from past events? | Yes, partially, no | [14] | $RA_{1,3}$ |
| 17 | Have qualitative impairments of the water supply occurred in the past so that substitute supply measures were necessary? | Yes, partially, no | [12,22] | $RA_{1,4}$ |
| 18 | Have quantitative impairments of the water supply occurred in the past so that substitute supply measures were necessary? | Yes, partially, no | [12,22] | $RA_{1,5}$ |
| 19 | Does the hazard analysis include other potential hazards that have not yet occurred? | Yes, partially, no | [14,17] | $RA_{1,6}$ |
| 20 | Have hazards been identified that need to be prioritised? | Yes, partially, no | [14,17] | $RA_{1,7}$ |
| | Sub Indicator $RA_2$—Vulnerability Analysis | | | |
| 21 | Has a vulnerability analysis been conducted? | Yes, partially, no | [14,17] | $RA_{2,1}$ |
| 22 | Was the vulnerability analysis carried out in cooperation with water supply utilities and disaster management? | Yes, partially, no | [12,14] | $RA_{2,2}$ |
| 23 | Have scenarios been identified for the vulnerability analysis? | Yes, partially, no | [12,14] | $RA_{2,3}$ |
| 24 | Are the components to be analysed specified? | Yes, partially, no | [14] | $RA_{2,4}$ |
| 25 | Have you checked which components would be exposed to which hazards (exposure)? | Yes, partially, no | [14] | $RA_{2,5}$ |
| 26 | Has the functionality of the components been checked? | Yes, partially, no | [14] | $RA_{2,6}$ |
| 27 | Has the technical replaceability of the components been checked? | Yes, partially, no | [14] | $RA_{2,7}$ |
| 28 | Has the organizational replaceability of the components been checked? | Yes, partially, no | [14] | $RA_{2,8}$ |

**Table A3.** Survey sheet to assess the Emergency Preparedness Planning Indicator—Part 3.

| ID | Question | Reply Options | Source | Individual Indicator |
|---|---|---|---|---|
| | Main Indicator RA—Risk Analysis | | | |
| | Sub Indicator $RA_3$—Risk Identification | | | |
| 29 | Was the risk assessment carried out with the involvement of specialist authorities or research institutions? | Yes, partially, no | [14] | $RA_{3,1}$ |
| 30 | Has the extent of damage in the scenarios considered been determined? | Yes, partially, no | [14] | $RA_{3,2}$ |
| 31 | Was the assessment of the extent of damage carried out with the involvement of those responsible for civil protection in the county/city? | Yes, partially, no | [14] | $RA_{3,3}$ |
| 32 | Was the probability of occurrence determined in the scenarios considered? | Yes, partially, no | [14,21,45] | $RA_{3,4}$ |
| 33 | Was the probability of occurrence carried out with the involvement of specialist authorities or research institutions? | Yes, partially, no | [16] | $RA_{3,5}$ |
| 34 | Was the number of inhabitants affected in the scenarios considered determined? | Yes, partially, no | [14] | $RA_{3,6}$ |
| 35 | Has the probability of occurrence and the extent of damage been classified on a scale (e.g., according to [14])? | Yes, partially, no | [14] | $RA_{3,7}$ |
| | Sub Indicator $RA_4$—Risk Comparison and Evaluation | | | |
| 36 | Were the scenarios compared using a risk matrix? | Yes, partially, no | [14] | $RA_{4,1}$ |
| 37 | Were the scenarios prioritized using a risk matrix? | Yes, partially, no | [14] | $RA_{4,2}$ |

**Table A4.** Survey sheet to assess the Emergency Preparedness Planning Indicator—Part 4.

| ID | Question | Reply Options | Source | Individual Indicator |
|---|---|---|---|---|
| | Main Indicator PM—Preventive Measures | | | |
| | Sub Indicator $PM_1$—Structural Redundancies | | | |
| 38 | Are the extraction plants designed redundantly? | Yes, partially, no | [14] | $PM_{1,1}$ |
| 39 | Are the storage tanks designed redundantly? | Yes, partially, no | [17] | $PM_{1,2}$ |
| 39.1 | quantitatively redundant? | Yes, partially, no | [14] | $PM_{1,2,1}$ |
| 39.2 | structurally redundant? | Yes, partially, no | [14] | $PM_{1,2,2}$ |
| | Sub Indicator $PM_2$—Interrelation of Supply | | | |
| 40 | Are there supply links with other water utilities? | Yes, partially, no | [14,26] | $PM_{2,1}$ |
| | Sub Indicator $PM_3$—Grid Construction | | | |
| 41 | Are there supply links with other water utilities? | Yes, partially, no | [17,28] | $PM_{3,1}$ |
| 42 | Have grid development measures, which are necessary to ensure security of supply, been implemented? | Yes, partially, no | [27] | $PM_{3,2}$ |
| | Sub Indicator $PM_4$—Remote Monitoring, Control Systems | | | |
| 43 | Is the supply system connected to a remote monitoring system? | Yes, partially, no | [17,29,30] | $PM_{4,1}$ |
| 44 | Is the supply system equipped with a state-of-the-art control system? | Yes, partially, no | [14] | $PM_{4,2}$ |
| | Sub Indicator $PM_5$—General Measures | | | |
| 45 | Have renewal measures necessary to ensure security of supply been implemented? | Yes, partially, no | [17,32] | $PM_{5,1}$ |
| 46 | Have maintenance measures necessary to ensure security of supply been implemented? | Yes, partially, no | [17,32] | $PM_{5,2}$ |
| 47 | Have physical protection measures, which are necessary to ensure security of supply, been implemented? | Yes, partially, no | [14,31] | $PM_{5,3}$ |

**Table A5.** Survey sheet to assess the Emergency Preparedness Planning Indicator—Part 5.

| ID | Question | Reply Options | Source | Individual Indicator |
|---|---|---|---|---|
| | Main Indicator CM—Crisis Management | | | |
| | Sub Indicator $CM_1$—Organisation and Coordination | | | |
| 48 | Does the water utility develop contingency plans in addition to the action plans according to the Drinking Water Ordinance? | Yes, partially, no | [14,35] | $CM_{1,1}$ |
| 49 | Are contingency plans for emergency situations in the water supply developed by the civil protection authority? | Yes, partially, no | [14,35] | $CM_{1,2}$ |
| 50 | Are you familiar with the content of these plans? | Yes, partially, no | [14,34] | $CM_{1,2}$ |
| 51 | Are the contact details of the following contact persons known? | Yes, partially, no | | $CM_{1,3}$ |
| 51.1 | Water utility | Yes, no | [17,23] | $CM_{1,3,1}$ |
| 51.2 | Emergency management / civil protection | Yes, no | [17,23] | $CM_{1,3,2}$ |
| 51.3 | Principal administrator | Yes, no | [14] | $CM_{1,3,3}$ |
| 51.4 | Fire department | Yes, no | [12] | $CM_{1,3,4}$ |
| 51.5 | Federal Agency for Technical Relief | Yes, no | [12] | $CM_{1,3,5}$ |
| 51.6 | Red Cross | Yes, no | [12] | $CM_{1,3,6}$ |
| 51.7 | Civil-Military Cooperation | Yes, no | [12] | $CM_{1,3,7}$ |
| 51.8 | National Command | Yes, no | [12] | $CM_{1,3,8}$ |
| 51.9 | Other authorities (e.g., health, environment, police) | Yes, no | [12] | $CM_{1,3,9}$ |
| 51.10 | Press | Yes, no | [12] | $CM_{1,3,10}$ |
| 52 | Are sensitive facilities available in the supply area? | | [14] | $CM_{1,4}$ |
| 52.1 | Hospital | Yes, no | | $CM_{1,4,1}$ |
| 52.2 | Nursing home | Yes, no | | $CM_{1,4,2}$ |
| 52.3 | Kindergarten/School | Yes, no | | $CM_{1,4,3}$ |
| 52.4 | Dialysis centers | Yes, no | | $CM_{1,4,4}$ |
| 53 | Are sensitive facilities in the supply area included? | | [14] | $CM_{1,5}$ |
| 53.1 | Hospital | Yes, partially, no | | $CM_{1,5,1}$ |
| 53.2 | Nursing home | Yes, partially, no | | $CM_{1,5,2}$ |
| 53.3 | Kindergarten/School | Yes, partially, no | | $CM_{1,5,3}$ |
| 53.4 | Dialysis centers | Yes, partially, no | | $CM_{1,5,4}$ |
| 54 | Are the telephone number, location and capacity of the following facilities in the district/district free city recorded? | | [12] | $CM_{1,6}$ |
| 54.1 | Brewery/beverage manufacturer | Yes, partially, no | | $CM_{1,6,1}$ |
| 54.1.1 | telephone number | Yes, partially, no | | $CM_{1,6,1,1}$ |
| 54.1.2 | location | Yes, partially, no | | $CM_{1,6,1,2}$ |
| 54.1.3 | capacity | Yes, partially, no | | $CM_{1,6,1,3}$ |
| 54.2 | Beverage suppliers (beverage market) | | | $CM_{1,6,2}$ |
| 54.2.1 | telephone number | Yes, partially, no | | $CM_{1,6,2,1}$ |
| 54.2.2 | location | Yes, partially, no | | $CM_{1,6,2,2}$ |
| 54.2.3 | capacity | Yes, partially, no | | $CM_{1,6,2,3}$ |
| 54.3 | Carriers | | | $CM_{1,6,3}$ |
| 54.3.1 | telephone number | Yes, partially, no | | $CM_{1,6,3,1}$ |
| 54.3.2 | location | Yes, partially, no | | $CM_{1,6,3,2}$ |
| 54.3.3 | capacity | Yes, partially, no | | $CM_{1,6,3,3}$ |
| 54.4 | Neighbouring Utilities | | | $CM_{1,6,4}$ |
| 54.4.1 | telephone number | Yes, partially, no | | $CM_{1,6,4,1}$ |
| 54.4.2 | location | Yes, partially, no | | $CM_{1,6,4,2}$ |
| 54.4.3 | capacity | Yes, partially, no | | $CM_{1,6,4,3}$ |

**Table A6.** Survey sheet to assess the Emergency Preparedness Planning Indicator—Part 6.

| ID | Question | Reply Options | Source | Individual Indicator |
|----|----------|---------------|--------|---------------------|
| | Main Indicator CM—Crisis Management | | | |
| | Sub Indicator $CM_2$—Provision of Resources | | | |
| 55 | Are the resources required for the substitute water supply (mobile treatment plants, mobile pipelines, transport vehicles) kept available or is access to them ensured? | Yes, partially, no | [14] | $CM_{2,1}$ |
| 56 | Are the materials required for the substitute water supply (pressure increasing systems, hose connections, etc.) kept in stock or is access ensured? | Yes, partially, no | [14] | $CM_{2,2}$ |
| | Sub Indicator $CM_3$—Exercises | | | |
| 57 | Was the interaction with the authorities and organisations involved in the event of a crisis in the water supply discussed? | Yes, partially, no | [14,17,23] | $CM_{3,1}$ |
| 58 | Has the interaction with the authorities and organisations involved been practised for crisis situations in the water supply? | Yes, partially, no | [14,17,23] | $CM_{3,2}$ |
| | Sub Indicator $CM_4$—Communication | | | |
| 59 | Is access to communication media ensured in the event of a crisis? | Yes, partially, no | [14,17,36] | $CM_{4,1}$ |

**Table A7.** Survey sheet to assess the Emergency Preparedness Planning Indicator—Part 7.

| ID | Question | Reply Options | Source | Individual Indicator |
|----|----------|---------------|--------|----------------------|
| | Main Indicator E—Evaluation | | | |
| | Sub Indicator $E_1$—Evaluation Preliminary Planning | | | |
| 60 | Are the awareness raising aspects regularly evaluated? | Yes, partially, no | [14,17] | $E_{1,1}$ |
| 61 | Are the definitions of responsibilities regularly evaluated? | Yes, partially, no | [14,17] | $E_{1,2}$ |
| | Sub Indicator $E_2$—Evaluation Risk Analysis | | | |
| 62 | Is the hazard analysis regularly evaluated? | Yes, partially, no | [14,17] | $E_{2,1}$ |
| 63 | Is the vulnerability analysis regularly evaluated? | Yes, partially, no | [14,17] | $E_{2,2}$ |
| 64 | Is the risk identification regularly evaluated? | Yes, partially, no | [14,17] | $E_{2,3}$ |
| 65 | Are the risk comparison and evaluation regularly evaluated? | Yes, partially, no | [14,17] | $E_{2,4}$ |
| | Sub Indicator $E_3$—Evaluation Preventive Measures | | | |
| 66 | Are the structural redundancies regularly evaluated? | Yes, partially, no | [14,17] | $E_{3,1}$ |
| 67 | Are the interrelations of supply regularly evaluated? | Yes, partially, no | [14,17] | $E_{3,2}$ |
| 68 | Are the grid construction measures regularly evaluated? | Yes, partially, no | [14,17] | $E_{3,3}$ |
| 69 | Are the remote monitoring and control systems regularly evaluated? | Yes, partially, no | [14,17] | $E_{3,4}$ |
| 70 | Are the general measures regularly evaluated? | Yes, partially, no | [14,17] | $E_{3,5}$ |
| | Sub Indicator $E_4$—Evaluation Crisis Management | | | |
| 71 | Are the organisation and coordination measures regularly evaluated? | Yes, partially, no | [14,17] | $E_{4,1}$ |
| 72 | Is the provision of resources regularly evaluated? | Yes, partially, no | [14,17] | $E_{4,2}$ |
| 73 | Are the exercised regularly evaluated? | Yes, partially, no | [14,17] | $E_{4,3}$ |
| 74 | Are the communication measures regularly evaluated? | Yes, partially, no | [14,17] | $E_{4,4}$ |

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
