# Peer review of "Batten Down the Hatches—Assessing the Status of Emergency Preparedness Planning in the German Water Supply Sector with Statistical and Expert-Based Weighting"

_sustainability, doi:10.3390/su12177177_

Round 1

Reviewer 1 Report

The article presents its findings in the very clean manner – from methodology, data used, to discussion and limitations of the study. I must say, that was little bit surprised by similarity in the results for both weighting methods – but the result is well argued and seems to be correct based on presented data.

I recommend publishing the article after some minor edits as specified bellow:

  • Page 5 – table 1 – consider revising it. The column composite indicator serves no purpose in the table (the information is already present in the table label), removing it will allow to present better remaining data.
  • Page 6 row 92 consider revisiting sentence … and characterized by a certain security relevance. What exactly is meant by that? Is the information classified?
  • Page 8 row 171 – p < 05, should be probably p < 0.05 or p < .05
  • Page 8 figure 4 – at first I thought, that the figure is missing, but it is only very small – resizing is needed
  • Page 9 row 205 and page 10 row 213 (p >> 0.05) the >> symbol was probably replaced by automatic correction
  • Page 12 row 235 in sentence … for self-assessment of emergency preparedness planning and resilience … I recommend removing the word resilience, as the article is not focused on it. The resilience is different concept requiring for example studying ability to bounce back of the system. Removing resilience does not detract from article’s findings.
  • Page 13 row 270 (2) should be instead of ()2

Reviewer 2 Report

This is an interesting study. My suggestions for improvement are directed mainly towards improving the flow and presentation of the paper.

  1. Please have a native English speaker review the manuscript for minor edits related to grammar.
  2. I would suggest different graphical depictions of the results. Bar charts are excellent however figures 3, 6, 8, 9 are difficult to read, interpret and provide very little value. How can this be presented better for the reader to understand the difference of indicators. For indicators to be useful they must be able to present data in a way that is understandable to lay users. 
  3. Materials and Methods needs further clarity and explanation. For example line item 72 is not clear ..."underlying main, partial and individual organisational and technical indicators. " explain this further please
  4. Explain this line item 46-47 this seems confusing Figure 1 shows the five steps of risk and crisis
    47 management according to [12–14].
